# Genetic Diversity and Relationships Among Tunisian Wild and Cultivated *Rosa* L. Species

**DOI:** 10.3390/plants13243563

**Published:** 2024-12-20

**Authors:** Khouloud Chtourou, Juan Alfonso Salazar, Germán Ortuño-Hernández, Najla Mezghani, Neila Trifi-Farah, Pedro Martínez-Gómez, Lamia Krichen

**Affiliations:** 1LR99ES12, Laboratoire de Génétique Moléculaire, Immunologie et Biotechnologie, Faculté des Sciences de Tunis, Université de Tunis El Manar, Tunis 2092, Tunisia; khouloud.chtourou@etudiant-fst.utm.tn (K.C.); neila.trifi@fst.utm.tn (N.T.-F.); lamia.krichen@issbat.utm.tn (L.K.); 2Department of Plant Breeding, Centro de Edafología y Biología Aplicada del Segura—Consejo Superior de Investigaciones Científicas (CEBAS-CSIC), Campus Universitario Espinardo, E-30100 Murcia, Spain; gortuno@cebas.csic.es (G.O.-H.); pmartinez@cebas.csic.es (P.M.-G.); 3National Gene Bank of Tunisia, Boulevard Leader Yasser Arafat Z. I. Charguia 1, Tunis 1080, Tunisia; najla_mezghani@yahoo.fr; 4LR21AGR05, Research Laboratory, Management of Horticultural Species in Organic and Conventional System, High Agronomic Institute of Chott Mariem, University of Sousse, Sousse 4042, Tunisia

**Keywords:** *Rosa* spp., microsatellite primers, germplasm, crop evolution, genetic diversity

## Abstract

Assessing and determining genetic diversity in rose species is a crucial step for conservation efforts, the establishment of a core collection, and the development of new varieties. This study represents the first investigation of genetic diversity among various rose species at different ploidy levels in Tunisia, with the aim of elucidating the genetic structure of the *Rosa* genus. It encompasses both spontaneous and cultivated accessions, featuring local and introduced species recognized for their adaptability, ornamental value, and fragrance. A total of 114 accessions representing eight rose species were collected. Significant genetic diversity was assessed using seven SSR markers, yielding an average of 21 alleles per locus and a PIC value ranging from 0.882 to 0.941. The results identified 343 phenotypic alleles across the seven primers, with 72 for the primer RhE2b (LG6), 55 for H10D03 (LG7), and 54 for RhB303 (LG2). One key finding was that most perfumed rose accessions (*R. damascena* Mill. and *R. centifolia* L.) are distinct from the other rose accessions, indicating a unique genetic pool for these roses. Another important finding was that the Tunisian accessions of ‘Rose of Ariana’ were closely related to *R. centifolia* accessions, commonly known as the ‘Rose of May’ from Grasse, France. To clarify the phylogeny of this species or subspecies, further molecular studies are warranted. Additionally, nomenclature confusion was identified between *R. sempervirens* L. and *R. canina* L. in the northwestern region of Tunisia, indicating that all wild accessions correspond to *R. sempervirens*.

## 1. Introduction

Rose is a remarkable species renowned for its wonderful visual appearance, fragrant perfume, noble uses, and diverse applications in cosmetics, gastronomy, phytotherapy, and perfume due to its numerous active components, making it inherently intriguing to study. It has significant potential for use in garden decoration, whether as a climbing plant, as a potted plant, or for enhancing flower beds. Additionally, it is a valuable source for essential oils and rose water production, and is widely used in the food and cosmetics industries. Its potential also extends to cut roses, including tea hybrids, which are supported by the development of the associated industry, elevating the rose as a prestigious and noble symbol of celebrating joyous occasions and expressing emotions.

The rose belongs to the *Rosaceae* family and the *Rosa* genus. Wild rose species are primarily native to Asia and Europe, with some specimens found in North America and North Africa [1,2]. The *Rosa* genus encompasses more than 200 species and 30,000 cultivars, a diversity attributed to interspecific hybridization. Despite this vast array, only ten rose species (*R. chinensis* Jacquin., *R. foetida* Herm., *R. gallica* L., *R. gigantea* Collet., *R. moschata* Herrm., *R. multiflora* Thunb., *R. phoenicia* Boiss., *R. rugosa* Thumb., and *R. wichurana* Crep.) have significantly contributed to the development of cultivated roses [3,4,5]. In the Tunisian flora described by Pottier-Alapetite (1979) [6], eight wild species were growing in Tunisia: *R. gallica* L., *R. semperviens* L., *R. canina* L., *R. moschata* Herrm., *R. agrestis* Savi., *R. micrantha* Sm., *R. stylosa* Desv., and *R. sicula* Tratt..

For several years, research work on roses in Tunisia has focused on characterizing the wild species *R. agrestis*, *R. semperviens*, *R. canina*, *R. moschata*, and *R. micrantha*, analyzing the morphometric character variation [7] and the biochemical parameters [8,9]. Despite morphological and biochemical markers, very limited work on the valorization of rose species and varieties of essential oils and extracted flower water have been conducted. No study has been conducted yet on the overall genetic diversity or identification of local germplasm using molecular markers, except for the 14 genotypes of the ‘Rose of Ariana’ from Ariana city, preserved in the National Gene Bank (NGBT) [10].

Tunisia is known for its good quality of fragrant rose essential oil and floral water extracted from *R. damascena*. The essential oil of *R. damascena* is a complex mixture of various compounds, particularly geraniol, citronellol, and nerol. These compounds have demonstrated beneficial effects in the food industry by preventing spoilage, as well as exhibiting antioxidant, antifungal, and antibacterial properties [11,12]. Additionally, rose essential oil is recognized for its antiseptic, antidepressant, astringent, digestive, antiallergic, anticephalic, and antimigraine effects, as well as its potential benefits for treating epilepsy and supporting reproductive health [13,14,15]. Dried rose petals are commonly added to yogurt as a flavoring agent, and this combination has been shown to alleviate digestive system disorders [16]. Nevertheless, other rose species of significant importance, such as ‘Rose of Ariana’*, R. moschata*, and *R. canina*, require further study and valorization. ‘Rose of Ariana’, once a symbolic ornamental plant of Ariana city, adorned roundabouts and public places over 30 years ago. Although, it holds significant cultural and environmental value, ‘Rose of Ariana’ is now under threat of disappearing. It is no longer widely recognized by citizens and rarely planted in green parks [17].

Despite the global economic value of rose species, Tunisia’s biodiversity is facing increasing threats from climate change, a shift towards more profitable crops, and urban expansion, leading to heightened genetic erosion and a greater risk of extinction for certain *Rosa* species in the northern and central regions. This underscores the urgent need to protect and conserve these populations. For cultivated roses in Tunisia, including those grown for cut flowers, nursery propagation as ornamental garden plants, and fragrance production, these areas have become increasingly widespread and important in recent years. For the Kairouan region, there has been an expansion in the production of *R. damascena*, reaching 920 hectares and 800 tons of roses in 2023 [18].

Characterization and assessment of diversity through molecular markers are essential prerequisites for understanding the genetic relationships among rose cultivars and their phylogeny. This is crucial for optimizing the utilization of valuable genetic attributes, preserving diverse germplasm collections, and strategically planning any breeding program [19,20,21,22]. The *Rosa* genus exhibits a fundamental chromosomal number of n = 7 [23], with ploidy levels ranging from diploid (2n = 2x = 14) to decaploid (2n = 10x = 70) across various wild species and cultivars [24]. These characteristics contribute to the complexity of studying genetic relationships within the *Rosa* genus and the ongoing debate surrounding the classification of cultivated and botanical roses.

Microsatellite or simple sequence repeat (SSR) markers are ideal tools for evaluating genetic variability among rose species and varieties [25], and are valuable tools for elucidating taxonomic relationships and nomenclature confusions within the *Rosa* genus [26,27]. They are suitable for estimating genetic variation among rose species, differentiating between wild and cultivated *Rosa* species, and analyzing modern garden rose cultivars [28,29,30,31].

In this work, we aim to identify the extent and structure of genetic diversity using microsatellite markers. This includes resolving issues of nomenclature confusion and clarifying the identity and phylogeny of the ‘Rose of Ariana’. Ultimately, this effort will lead to the selection and establishment of a core rose collection that represents the identified diversity.

## 2. Results

### 2.1. Genetic Diversity Analysis and Polymorphism Level Identified Among Rosa Species in Tunisia

All of the seven SSR primers produced scorable bands (Appendix A) and displayed polymorphism among the 114 studied genotypes, generating a total of 147 different alleles, with an average of 21 alleles per marker. Allele size ranged from 119 bp (RhB303) to 292 bp (RW10M24). The highest number of alleles (23) was observed for the primers RhE2b and RW52D4, and the lowest number of alleles (17) was observed for H20D08. The 147 identified alleles generated a total of 343 allelic phenotypes among the studied rose accessions. The primer RhE2b (LG6) showed the highest discriminating power (63.16%), with 72 different allelic phenotypes, followed by H10D03 (LG7) (55 allelic phenotypes) and RhB303 (LG2) (54 allelic phenotypes) (Table 1).

Observed heterozygosity (Ho) is correlated with the number of alleles. The highest level of Ho (0.706) was recorded for RW52D04, while the lowest value (0.128) was noted for H20D08. For expected heterozygosity (He), the lowest value was observed for RhB303 (0.803), while the highest value was noted for H10D03 (0.924), resulting in an average of 0.882, which is higher than Ho (0.483). These results suggest an excess of heterozygosity due to the natural interspecific hybridization characteristic of the *Rosa* germplasm.

The polymorphism information content (PIC) for the genotypes analyzed ranged from 0.881 to 0.941. The highest PIC value was observed for RhE2b (0.941), followed by H10D03 (0.927), while the lowest PIC value was observed for RhB303 (0.882), with an average value of 0.908 among the seven SSR markers. These high levels of PIC reveal the efficiency of the markers used in detecting polymorphism in the studied genetic material.

The AMOVA performed on 114 individuals at the threshold for statistical significance level of α = 1% indicated that the differentiation between groups was statistically significant (*p* = 0.001) (Table 2). The variation was partitioned between the within-group variability (89%) and the among-group variability (11%). The high within-group variability is a reminder that genetically distinct individuals correspond to the genetic group, and the significant intergroup variability suggests a limited gene flow between them.

### 2.2. Genetic Diversity Structuration Among Rosa Species in Tunisia

Our study investigates the genetic diversity of rose species in Tunisia, including modern roses (e.g., Modern English roses from the David Austin collection, tea hybrids, wichurana hybrids) and fragrant species (*R. damascena*, *R. centifolia*, *R. bourboniana* Desp., ‘Rose of Ariana’).

Using SSR molecular markers, genetic diversity structuration among *Rosa* species in Tunisia was assessed via a hierarchical ascending classification (HAC) constructed using Bruvo genetic distances [32] and the hierarchical classification method of Ward [33]. Bruvo genetic distances ranged from 0 for the identical accessions of *R. centifolia* RCe (1), (2), (3), and (4); *R. damascena* RD (1), (2), and (3) from the Agricultural Development Group (GDA) Sidi Amor collection; ‘Rose of Kairouan’ RK.IB ((1) and (3)); and RCe.IB ((1) and (2)), to 0.985 (the highest distance observed between the modern rose Annapurna (A.SA) and *R. damascena* (RD.S(4)).

HAC revealed a clear genetic structure in two major groups, I and II, and helped clarify the relationships among the studied accessions (Figure 1). The first cluster (I) regrouped accessions of the perfumed rose species (*R. damascena*, *R. centifolia* (RCe.IB), ‘Rose of Ariana’, and ‘Rose of Kairouan’ from Kairouan city). The second cluster (II) included accessions of *R. sempervirens* and *R. canina* from the northwest region of Tunisia, as well as *R. sempervirens* and the entire collection of modern roses from the GDA Sidi Amor.

Cluster I was subdivided into two sub-clusters (I.1 and I.2). Sub-cluster I.1 grouped ‘Rose of Ariana’ (except RA.SA) with *R. centifolia* (RCe.IB), and sub-cluster I.2 included the accessions of *R. damascena* from the Sfax region and from GDA Sidi Amor, as well as ‘Rose of Kairouan’ accessions from Kairouan region, suggesting a common ancestral origin despite the distance between the geographic origin of the accessions.

Cluster II was divided into two sub-clusters (II.1 and II.2). Sub-cluster II.1 included all the accessions of *R. sempervirens* and *R. canina* from the northwest region of Tunisia, as well as *R. sempervirens* from GDA Sidi Amor (RS.SA). *R. moschata* (RM.SA) from GDA Sidi Amor and its hybrids were grouped with the entire collection of modern roses in sub-cluster II.2.

The 75 garden rose cultivars studied in this work belong to ten different types: floribunda, hybrid tea, hybrid moschata, hybrid wichurana, shrubs, polyantha, Portland, climber, chinensis, and English modern rose by David Austin.

Concerning the modern English roses studied, as classified by David Austin (2012) [34], they are divided into three distinct groups. The first group consists of hybrid old roses, also known as the original English roses, characterized by their pink, crimson, or purple flower colors and a strong fragrance. Notable examples include ‘Brother Cadfael’, ‘Gertrude Jekyll’, ‘Hyde Hall’, ‘L.D. Braithwaite’, ‘Sharifa Asma’, ‘The Mayflower’, ‘Susan Williams-Ellis’, and ‘Mary Rose’. The second group encompasses English moschata hybrids, known for their sweet fragrance and varied flower colors; this group includes ‘Charlotte’, ‘Graham Thomas’, ‘Lady Emma Hamilton’, and ‘Molineux’. The Leander English roses represent a third group, which has expanded the range of flower colors while maintaining a robust fragrance, with varieties such as ‘Benjamin Britten’, ‘Golden Celebration’, ‘Grace’, ‘Teasing Georgia’, ‘The Alnwick Rose’, and ‘Leander’. Lastly, a collection of modern English cultivars does not fit neatly into the previous subgroups, including ‘Eustacia Vye’, ‘Lady of Shalott’, ‘Princess Alexandra of Kent’, and ‘The Pilgrim’.

It is noteworthy to highlight the presence of samples of fragrant roses, including *R. centifolia* from GDA Sidi Amor (RCe.SA (1), (2), (3), and (4)), ‘Almadinah rose’ (RMé.SA), the bourbon rose ‘Madame Isaac Pereire’ (RB.MIP.SA), and the old fragrant rose (OR.LK), as well as ‘Rose of Ariana’ (RA.SA), the accession of ‘Rose de Rescht’ (RdR.SA), and the accession of *R. gallica*—‘Rose de Provins’ (RG.RdP.SA) from GDA Sidi Amor, along with the three accessions of ‘Rose of Kairouan’ from the private garden in Ariana city (RK.IB), included in subgroup II.2.

To reinforce the observed diversity structure and validate the grouping of accessions, we employed two additional methods: a change in variables in a 3D space using principal component analysis (PCA) and a model-based Bayesian clustering method with STRUCTURE.

The first plan of the PCA (defined by the first two components, accounting for 46.4% of the total variance) revealed a structure that aligned well with the structure observed with HAC analysis (Figure 2). The studied genotypes were clustered into three major groups. Cluster (A) included perfumed roses, including *R. damascena*, *R. centifolia*, ‘Rose of Ariana’, *R. gallica*—‘Rose de Provins’ (RG.RdP.SA), and hybrid moschata ‘Sidi Amor’ (HM.SA.SA). Cluster (B) comprised modern English roses by David Austin and almost all hybrid roses. In cluster (C), RS.SA were grouped alongside the accessions of *R. sempervirens* from the northwest region of Tunisia, as well as the three accessions of *R. canina* (RC.S, RC.B(1), and RC.B(2)) with the three *R. moschata* hybrids (‘Felicia’ (HM.F.SA), ‘Ballerina’ (HM.B.SA), and ‘Clérance et Rosalie’ (HM.CR.SA)). This cluster also included the accessions of ‘Hyde Hall’ (HH.SA), ‘Indica Major’ (IM.SA), and ‘The Fairy’ (TF.SA). Finally, according to the first PCA plan, the modern rose type was characterized by the largest gene pool, though with likely lower intra-group variability, despite having a larger sample size (Table 2).

The model-based Bayesian clustering approach, implemented with the STRUCTURE software v2.3.4, was utilized to better understand the genetic structure among the 114 accessions of the *Rosa* genus in Tunisia (Figure 3). The rate of change in log-likelihood values between successive analyses (ΔK) indicated that K = 3 is the optimal level for clustering. This model-based approach leverages genetic information to determine the population membership of individual genotypes without assuming predefined populations [35]. The fragrant rose genotypes (*R. centifolia* (RC.IB (1), (2) and (3)), *R. damascena*, ‘Rose of Ariana’ (except RA.SA), and *R. gallica*—‘Rose de Provins’) were grouped in cluster I (red). Cluster II (blue) included the wild rose *R. sempervirens* and *R. canina* from the northwest of Tunisia and RS.SA, which exhibits higher genetic similarity with the fragrant accessions RB.MIP.SA and the four accessions of RCe.SA. In cluster III (green), RMé.SA and RA.SA, both from the fragrant rose group, were classified within the modern rose accessions.

### 2.3. Resolving the Phylogeny of the ‘Rose of Ariana’ and the Nomenclature Confusions of Rose Species in Tunisia

The key finding from the described results is that (1) almost all the fragrant roses were grouped in the same cluster, according to HAC and PCA, and (2) the hypothesis regarding the identity of ‘Rose of Ariana’ as an *R. centifolia* species is the most probable. To confirm this, we decided to explore this group of fragrant species in greater detail. Thus, HAC and Bayesian clustering analysis were assessed based on the accessions of *R. damascena*, *R. centifolia*, *R. gallica*—‘Rose de Provins’ (RG.RdP.SA), *R. moschata* (and its hybrids), *R. sempervirens*, ‘Rose of Kairouan’, ‘Rose of Ariana’ (RA.SA), ‘Almadinah rose’ (RMé.SA), the bourbon rose ‘Madame Isaac Pereire’ (RB.MIP.SA), ‘Rose de Rescht’ (RdR.SA), and the old fragrant rose (OR.LK).

HAC showed that the group of fragrance roses was divided into two major groups, I and II. The first group (I) was subdivided into subgroup I.1, including the accessions of ‘Rose of Ariana’ from different regions (NGBT and Ariana city gardens) and the accessions of *R. centifolia* from the Ariana city gardens (RCe.IB), and subgroup I.2, including the accessions of *R. damascena* from the three regions of Sfax, Kairouan, and GDA Sidi Amor, as well as the two accessions of ‘Rose of Ariana’ (RA.LK (3) and RA.LK (4) from Ariana city garden) that were grouped with the modern hybrid of *R. moschata* Sidi Amor (HM.SA.SA).

The second group (II) was divided into two subgroups. Subgroup II.1 included accessions of wild roses, such as *R. sempervirens* and *R. canina*, as well as hybrids of *R. moschata*, specifically, ‘Eclats d’Ambre’ (HM.EdA.SA) and ‘Felicia’ (HM.F.SA). Subgroup II.2 included the four accessions of *R. centifolia* (RCe.SA), which were related to the bourbon roses (RB.MIP.SA, OR.LK, RMé.SA). The *R. moschata* hybrid ‘Buff Beauty’ (HM.BB.SA) and the accessions of the ‘Rose of Kairouan’ (RK.IB) were grouped together, along with the *R. moschata* hybrids ‘Buff Beauty’ (HM.BB.SA) and ‘Ballerina’ (HM.B.SA), RA.SA, RG.RdP.SA, RdR.SA, and the two wild roses *R. moschata* (RM.SA) and *R. banksiae* Ait. (RC.B.SA) (Figure 4).

In the model-based Bayesian clustering method with STRUCTURE analysis (Figure 5), for the 58 fragrant rose genotypes, the change rate in the log-likelihood between successive K values (ΔK) revealed that the most probable Evanno coefficient after 20 iterations of structuring is K = 6. The first cluster (red) comprised accessions of *R. centifolia* (RCe.IB), along with two accessions of ‘Rose of Ariana’ (RA.LK (3) and RA.LK (4)) and a hybrid of *R. moschata* (HM.SA.SA) and RD.S(1). The second cluster (green) included all accessions of *R. damascena* (from GDA Sidi Amor and Sfax), as well as accessions of ‘Rose of Kairouan’ from Kairouan. The third cluster (blue) encompassed RS.SA, all accessions of *R. sempervirens*, the two accessions of *R. canina* (RC.B (1) and RC.B(2)) from the northwestern region of Tunisia, the accession of *R. canina* from the Sfax region (RC.S), and four accessions of *R. centifolia* from GDA Sidi Amor and the bourbon rose RB.MIP.SA. The fourth (yellow) and the fifth (violet) cluster includes the remaining accessions of ‘Rose of Ariana’ and the three accessions of RK.IB, respectively. The last cluster (light blue) is characterized by accessions of *R. moschata* and its hybrids (except HM.SA.SA), the wild rose HC.B.SA, and the remaining fragrant rose accessions (RdR.SA, RA.SA, RMé.SA, OR.SA, and RG.RdP.SA (40% of assignment rate).

To clarify the phylogenetic relationships, we examined the allelic profiles of the ‘Rose of Ariana’ accessions. We observed that for the seven SSR markers, this accession shared 12 alleles combine with 16 with the accessions of *R. centifolia* (RCe.IB), which itself shares alleles with *R. gallica*—‘Rose de Provins’ (Table 3).

Additionally, the four accessions of *R. centifolia* from GDA Sidi Amor did not cluster with the other accessions of *R. centifolia* (RCe.IB) in group I, which we consider a reliable source of *R. centifolia*, and as indicated in Table 4, among the seven primers examined, the RCe.SA accessions exhibited identical profiles to that of the bourbon rose (RB.MIP.SA) for three primers (H10D03, RhE2b, and RW10M24) and shared two alleles combined for two other primers (RhB303 and RW52D04).

## 3. Discussion

In this study, we examined 114 rose accessions, including fragrant roses, modern roses of various types, and wild roses from different regions of Tunisia, using seven SSR primers.

Regarding the molecular analysis of genetic diversity, we identified 147 alleles across the seven markers examined, with 343 phenotypic alleles. Our results revealed an average number of 21 alleles, which is comparable to those reported by Vukosavljev et al. (2013) [31] (21.6 alleles), who analyzed the differentiation of 138 varieties of garden roses, and Gaurav et al. (2022) [28] (1.42 alleles), who studied the genetic diversity of 21 cultivated roses, including *R. damascena*, *R. wichuraiana*, *R. bourboniana*, and *R. moschata* using SSR markers. This indicates a higher degree of differentiation among individuals and a high level of polymorphism in our study. By using seven primers, we were able to effectively differentiate between the rose accessions, demonstrating that the roses in Tunisia exhibit significant molecular diversity. Furthermore, the PIC value varied from 0.881 for RhB303 to 0.941 for RhE2b, with an average of 0.908. This average is considerably higher than the mean PIC of 0.365 [28]. Among the markers, RhE2b (LG6) and H10D03 (LG7), followed by RhB303 (LG2), exhibited the highest numbers of allelic phenotypes (72, 55, and 54, respectively). These findings align with previous studies that reported a significant number of allelic phenotypes for RhE2b (62/138, 31/78, and 32/734) and RhB303 (58/138, 19/78, and 37/734) [31,36,37]. LG6 and LG2 are particularly notable for their high levels of genetic diversity, which are linked to the presence of QTLs on these linkage groups. Notably, a substantial number of QTLs on LG6 have been associated with anthocyanin, flavonol, and carotenoid contents in tetraploid roses [38].

Examining the HAC revealed that the accessions of perfumed rose species (*R. damascena*, *R. centifolia* (RCe.IB), ‘Rose of Ariana’ (except RA.SA), and ‘Rose of Kairouan’ from Kairouan city) were grouped in the same cluster (I) (Figure 1). Notably, the clustering of ‘Rose of Kairouan’ accessions from Dhraa Thammar, Khazzazia, and Raggeda in Kairouan with *R. damascena* accessions from Sfax and GDA Sidi Amor occurred despite their geographical separation. These findings are supported by Agaoglu et al. (2000) [39] and Baydar et al. (2004) [40], who studied the genetic diversity of *R. damascena* using RAPD and AFLP markers, respectively, and found genetic uniformity among these varieties from Turkey. Furthermore, Farooq et al. (2013) [41] identified a genetic association between Iranian and Pakistani accessions of *R. damascena*, grouping them with *R. centifolia*, which is explained by the fact that *R. centifolia* is a hybrid of *R. damascena* [42].

*R. damascena* accessions from Sfax (RD.S), GDA Sidi Amor (RD.SA), and ‘Rose of Kairouan’ accessions (from Khazzazia, Dhraa Tammar, and Raggeda) share the same gene pool, with 45.8% common alleles, suggesting a common ancestry. Across the seven markers, a total of 24 alleles, including both common and rare alleles, were identified. This indicates that the ‘Rose of Kairouan’ from Kairouan is not a distinct species endemic to this region but is derived from *R. damascena*. However, the ‘Rose of Kairouan’ accession collected from Ariana city (RK.IB) was found to differ, sharing only 19% of common alleles with *R. damascena* from Kairouan accessions. These results confirm that the *R. damascena* accessions from Kairouan, Sfax, and GDA Sidi Amor have the same genetic origin, with no significant differentiation between them. While this examination makes it possible to distinguish a variety specific to the Kairouan region within the same species, *R. damascena*, it is evident that, through epigenetic phenomena, this variety has developed adaptations to the region, its climate, and its cultivation practices, resulting in phenotypic modifications.

Looking at the HAC (Figure 1 and Figure 4) and Table 4, the three accessions of ‘Rose of Kairouan’ from Ariana city (RK.IB), along with the four *R. centifolia* accessions (RCe.SA), were in the same cluster (II) alongside modern rose accessions, including bourbon rose ‘Madame Isaac Pereire’ (RB.MIP.SA), ‘Almadinah rose’ (RMé.SA), and ‘old rose’ (OR.LK). This grouping is explained by the fact that these accessions share the same allelic phenotypes. This leads to the conclusion that there is nomenclature confusion, where the RCe.SA accessions do not correspond to true *R. centifolia*. Regarding ‘Rose of Kairouan’ (RK.IB), the analysis of the allelic profiles shows that the city of Kairouan is known for two distinct species: *R. damascena* (RK.R, RK.DT, and RK.Kh), cultivated by farmers in the regions of Khazzazia, Dhraa Thammar, and Raggeda, and a variety of *R. Bourboniana* (RK.IB), cultivated in Ariana. The study of the biochemical composition of these roses is important for understanding the genetic diversity of roses in Kairouan.

Focusing on sub-cluster II.2 (Figure 1), which includes almost all hybrids of roses and the modern English roses by David Austin (except HM.F.SA, HH.SA, and IM.SA), the roses were not grouped according to their species or hybrid status, unlike other groups. This result is consistent with the findings from the AMOVA test (Table 2). In fact, the low phiPT value (0.180) observed between rose groups indicates a lack of clear genetic differentiation. Our findings align with those of Vukosavljev et al. (2013) [31], who reported that, although cultivars of each garden rose type generally clustered together, there was a significant overlap between the types. This continuous structure, with no clear separation between groups, was probably due to the importance of successive crossings between European rose hybrids and Asian roses, leading to genetically close roses. Several genetic groups summarized the information of intermediate steps during the hybridization process and were not truly isolated populations. The presence/absence of the coding of alleles may have led to a loss of information, amplifying this phenomenon [29]. According to the first PCA plan, the modern rose type was characterized by the largest gene pool, likely with lower intra-group variability, despite having a larger sample size (Table 2). This reduced variability can be attributed to the selection of specific phenotypic traits, such as floral architecture and fragrance, which led to phenotypic uniformity among modern English cultivars [31].

Based on the accessions of ‘Rose of Ariana’ and utilizing SSR markers along with a larger sample size of roses from various origins and species, our findings allowed the identity of ‘Rose of Ariana’ to be revealed. PCA (Figure 2) and HCA (Figure 1 and Figure 4) analyses demonstrated that all accessions of ‘Rose of Ariana’ were related to *R. centifolia* (RCe.IB), while *R. gallica*—‘Rose de Provins’ (RG.RdP.SA) was located in a different cluster. The structure of the diversity suggests that these accessions derive from *R. centifolia* or share closer genetic ties with this species, potentially due to historical hybridization events, or they could be an interspecific hybrid between *R. centifolia* and another *Rosa* sp. However, the model-based Bayesian clustering (Figure 3) revealed that the majority of ‘Rose of Ariana’ accessions (except RA.LK(3) and RA.LK(4)) were clustered with both *R. gallica*—‘Rose de Provins’ (RG.RdP.SA) and *R. centifolia* (RCe.IB). It is important to note that *R. gallica*—‘Rose de Provins’ (RG.RdP.SA), which is classified as *R. gallica officinalis* [43], exhibits significant differences from wild-type cultivars. Consequently, it could not be a representative accession for *R. gallica* in our study. The sharing of alleles between RCe.IB accessions and those of ‘Rose of Ariana’ (85.71% common alleles) raises the hypothesis that ‘Rose of Ariana’ is identified with *R. centifolia* or a mutant of *R. centifolia*. Its proximity to *R. centifolia* in hierarchical clustering analysis and principal component analysis suggests a shared lineage and inherited traits from its parent species. The independent cluster identified in the model-based Bayesian clustering analysis using STRUCTURE suggests a distinct genetic identity for ‘Rose of Ariana’, while the shared traits with *R. gallica*—‘Rose de Provins’ (40% assignment rate) imply a genetic contribution from this species. In this context, the proximity of ‘Rose of Ariana’ to *R. centifolia* in the phylogenetic tree suggests that *R. centifolia* is one of its parental species. To resolve the phylogenetic uncertainties surrounding the authenticity of ‘Rose of Ariana’, a comprehensive study of phylogeny based on chloroplast DNA and ancestral affiliation is necessary, along with an increase in the number of *R. gallica* accessions. This is important because RG.RdP.SA is not representative of wild *R. gallica*.

Focusing on the four accessions of ‘Rose of Ariana’ from Ariana city gardens, we found out that the accessions RA.LK (3) and RA.LK (4) were not clustered with the rest of the ‘Rose of Ariana’ accessions. In fact, the percentage of common alleles between RA.LK(3) and RA.LK(4), respectively, with the remaining ‘Rose of Ariana’ accessions corresponds to 43.75% and 31.25%, respectively, along with rare alleles for H10D03, H20D08, and RhB303 (Table 3). Indeed, the ‘Rose of Ariana’ accessions RA.LK (1), (2), (3), and (4) originated from the municipal nursery of the city of Ariana and were vegetatively propagated and sold during the Festival of Roses in Ariana, which is organized each year in May in the ‘Bir Belhassen’ rose gallery. In fact, each year, one accession of ‘Rose of Ariana’ (RA.LK1, 2, 3, 4) is bought and planted in the same garden. It was expected that they would be identical. Unfortunately, they were different enough to be in different groups of the HAC, showing the instability of the material sold during the Festival of Roses in Ariana city. This instability may be attributed to many hypotheses: (1) material originating from multiple maternal sources, (2) the instability of the ‘Rose of Ariana’ clones in the nursery, (3) the diversity of plant material origins, or (4) the genetic variation or mutations occurring in the maternal trees. This situation threatens the authenticity of the ‘Rose of Ariana’ identity and represents a risk of losing the original genetic profile of ‘Rose of Ariana’, highlighting the urgency for the municipality actors to remediate this confusion as soon as possible and for the nursery to standardize the vegetative propagation of ‘Rose of Ariana’ using stable and authentic clones.

Concerning the phylogeny of Tunisian *R. canina*, which is used for the extraction of floral waters and is locally known as ‘dog rose’, there is noted confusion regarding its identity—whether it is *R. canina* or *R. moschata*. PCA (Figure 2), HAC (Figure 1 and Figure 4), and the model-based Bayesian clustering method with STRUCTURE analysis (Figure 3 and Figure 5) revealed that all wild samples of *R. sempervirens* and *R. canina* were grouped together as *R. sempervirens* from GDA Sidi Amor (RS.SA). According to the model-based Bayesian clustering method, the accessions of *R. sempervirens* and the two accessions of *R. canina* from the northwestern region of Tunisia shared the same genetic pool with RS.SA, which appears to be a hybrid. A similar finding was observed with *R. canina* from Sfax. This suggests nomenclature confusion, indicating that these wild accessions could correspond to *R. sempervirens* accessions used for floral extraction. A comprehensive study in the Zaghouan region, the native area of ‘dog rose’, is necessary to clarify these identities based on a bigger sample data set.

## 4. Materials and Methods

### 4.1. Plant Material

Surveys were conducted in different areas to determine the distribution of the *Rosa* species across Tunisia. Wild rose species in Tunisia are predominantly localized along northwestern watercourses in Nefza (37°01′57.4″ N, 9°06′3.2″ E), Djebba (36°28′16.6″ N, 9°05′58.5″ E), and Tabbeba (36°54′54.2″ N, 9°09′41.8″ E and 36°54′45.6″ N, 9°08′77.2″ E) from region of Beja and Sejnane (37°04′8.44″ N, 9°16′23.97″ E) from region of Bizerte, where four accessions of *R. sempervirens* and two accessions of *R. canina* were collected, in addition to a modern rose accession (MR.B) from Beja (36°42′35.58″ N, 9°30′33.14″ E). Regarding fragrant roses, four accessions of *R. damascena* and one accession of *R. canina* were collected in the locations of Sfax city (34°47′45.22″ N, 10°45′31.01″ E) and Aouabed (34°50′36.8″ N, 10°38′30.1″ E) from the region of Sfax. In the region of Kairouan, prospections were carried out in the locations of Khazazia (35°37′5.85″ N, 10°11′42.51″ E), Dhraa Tammar (35°47′54.3″ N, 10°04′8.5″ E), and Raggeda (35°33′50.81″ N, 10°02′46.7″ E), known for the production of *R. damascena*, called ‘Rose of Kairouan’. From this region, three accessions were collected from each location. Furthermore, three accessions of *R. centifolia*, three accessions of ‘Rose of Kairouan’, and 11 accessions of ‘Rose of Ariana’, as well as an old fragrant rose (OR.LK), were collected from private gardens in Ariana city (36°51′25.49″ N, 10°11′0.53″ E and 36°52′21.09″ N, 10°10′43.09″ E) and from the National Gene Bank (NGBT) (36°50′28.45″ N, 10°12′41.04″ E). Prospections also concerned GDA Sidi Amor in Ariana (36°55′48.72″ N, 10°10′2.72″ E), where 75 rose accessions were selected from wild and modern species from a large collection of 200 varieties belonging to different species. Thus, a total of 114 rose accessions were collected from seven prospected regions representative of the rose cultivation areas in Tunisia (Figure 6) (Appendix A).

### 4.2. DNA Extraction and SSR Marker Amplification

Genomic DNA was extracted from young leaves of roses using a modified cetyl trimethyl ammonium bromide (CTAB) method [44] optimized to miniprep [45]. The quality of the DNA was visualized through electrophoresis on 0.8% agarose gels and quantified at 260 nm. The quality and the purity were measured at a 260/280 nm absorbance ratio using a NanoDrop One Spectrophotometer (Thermo Fisher Scientific, Waltham, MA, USA) [46]. The extracted DNA was then diluted to a concentration of 50 ng/μL and stored at −20 °C for polymerase chain reaction (PCR) use.

A total of seven SSR markers were selected to analyze the genetic diversity among the 114 collected accessions from different species, including wild, old, and modern varieties with varying levels of polyploidy (2×, 3×, 4×, and 5×). These seven SSR markers covered the entire genome of the *Rosa* genus (except LG3) and were chosen for their high level of polymorphism, independently of the ploidy level [19,47,48] (Table 5). The PCR was conducted in a reaction volume of 15 μL: 3 μL of Taq DNA buffer (5×), 1.2 μL of MgCl_2_ (25 Mm), 0.3 μL of dNTPs (10 mM), 0.3 μL each of forward and reverse primers (10 mM), 0.2 μL of Taq DNA polymerase (5 U/μL), 2.5 μL of sample DNA (50 ng/μL), and 7.2 μL Milli-Q water. The PCR reactions were carried out in a thermal cycler programmed at 94 °C for 5 min, followed by 35 cycles of denaturation at 94 °C for 30 s and annealing at 50–55 °C (annealing temperature was optimized for each primer) for 40 s and 72 °C for 30 s, with a final extension step of 72 °C for 10 min before cooling to standby at 4 °C. PCR products were resolved in a 3% MetaPhor agarose (Biowittaker, Rockland, ME, USA) gel stained with GelRedTM Nucleic Acid Gel Stain^®^ (Biotium, Fremont, CA, USA) using 1 Kb Plus DNA Ladder as a molecular size standard. The SSR profiles were visualized and captured using Amersham Imager 600 (GE Healthcare UK Limited, Buckinghamshire, UK) with UV fluorescence.

### 4.3. Data Analysis

Because of the polyploidy of rose species, it is difficult to consider the co-dominant scoring of the SSR markers in heterozygote samples; distinguish whether a particular allele was present in one, two, or three copies; and thus deduce the actual genotype of an accession. When a locus is analyzed as one character, we refer to this as the ‘allelic phenotype’ of the accession [47,49,50]. Thus, banding patterns observed at a particular locus were recorded as a presence/absence and were referred to as ‘allelic phenotypes’ [49]. Data are reported for a binary matrix as a presence (1) or absence (0) of bands at each polymorphic marker recorded among the seven SSR primers. A binary matrix was used for data analyses to determine the number of alleles per locus and the number of allelic phenotypes among all the genotypes. We determined the band sizes for each SSR marker using GelAnalyzer 23.1 software (GelAnalyzer 23.1, available at www.gelanalyzer.com (accessed on 14 January 2024), by Istvan Lazar, Jr., Ph D., and Istvan Lazar Sr., Ph D., CSc).

The polymorphic characteristics were calculated using SPAGeDi 1.3 [51]. The determined key parameters of genetic diversity correspond to the expected genetic heterozygosity (He), calculated as Equation (1), where *p_i_* is the frequency of the *i*th allele [52], representing the probability that two randomly chosen alleles at a locus within a set of genotypes will be different under Hardy–Weinberg equilibrium (i.e., assuming random mating) [31], and observed genetic heterozygosity (Ho), where Ho represents the number of heterozygous individuals divided by the total number of individuals, which measures population differentiation (genetic distance) based on allele frequency differences among populations [53].
He = 1 − ∑*p_i_*^2^(1)

The genetic structure analysis was assessed across the following data analyses: analysis of molecular variance (AMOVA) between genetic groups in empirical data was calculated with the Bruvo genetic distance [32] matrix using GenAlEx 6.503 [54,55]. This analysis divided the molecular variance among and within genetic groups on two levels. Pairwise PhiPT, an analog of Wright’s FST for dominant binary data, was also estimated with 1000 permutations [29,56].

Bruvo distances assume ambiguous allele copy numbers in partial heterozygotes and take mutational distances into account by including repeat themes of the microsatellites to consider the polyploidy of the studied genotypes. The conversion of allelic phenotypes into Bruvo distances was accomplished using POLYSAT software [57] within the R environment [58].

Polymorphism information content (PIC) was calculated as Equation (2), where *p_i_* and *p_j_* are allele frequencies at alleles *i* and *j*, respectively, and *n* is the number of alleles [59] using the Polysat package [57] running in the R environment.
(2)$$\mathrm{PIC}=1-(\sum_{i=1}^{n}p_{i}^{2})-\sum_{i=1}^{n-1}\sum_{j=i+1}^{n}2{p_{i}}^{2}{p_{j}}^{2}$$

Finally, for structure analysis, (1) principal component analysis (PCA) was assessed in R using the Stats package version 2.6.2 [60,61,62], (2) hierarchical tree classification (HTC) was assessed in R using Bruvo distances [32] and the Ward method of classification [33], and (3) the model-based Bayesian clustering method was performed on all rose accessions using the STRUCTURE 2.3.4 software [63]. This program assigns individual genotypes to distinct populations while identifying hybrid zones, migrants, and admixed individuals. STRUCTURE was executed for 20 replicates across each K value (ranging from 2 to 10), with a burn-in period of 100,000 iterations followed by 100,000 Markov chain Monte Carlo (MCMC) replications. For selecting the most likely number of clusters (K) supported by our dataset, we utilized the plateau criterion [63] and the ∆K method [64].

## 5. Conclusions

All the SSR primers produced scorable bands and showed polymorphism among the rose genotypes in Tunisia, generating a large diversity of distinct alleles. These alleles resulted in a wide range of allelic phenotypes among the rose accessions examined. HAC revealed a clear genetic structure into two major groups, clarifying the relationships among the studied accessions. The first group encompassed accessions of perfumed rose species (*R. damascena*, *R. centifolia*, and ‘Rose of Ariana’), while the second group included all accessions of *R. sempervirens* and *R. canina* from northwest Tunisia, as well as *R. sempervirens* from GDA Sidi Amor and the entire collection of modern roses located at GDA Sidi Amor. Our research elucidated the phylogeny of ‘Rose of Ariana’, identifying it as *R. centifolia*, commonly referred to as the ‘Rose of May’ or ‘Rose of Grasse’. However, the hypothesis that ‘Rose of Ariana’ is truly an *R. centifolia* or a hybrid between *R. centifolia* and another *Rosa* species remains unresolved. To accurately determine the phylogenetic relationships among accessions and species, extensive investigations using chloroplastic markers are necessary. The decline in wild species in various regions, exacerbated by climate change, underscores the urgent need to conserve the genetic diversity of rose species. Establishing a core collection for their conservation in Tunisia, in collaboration with gene banks, is essential for preserving this diversity.

## Figures and Tables

**Figure 1 plants-13-03563-f001:**
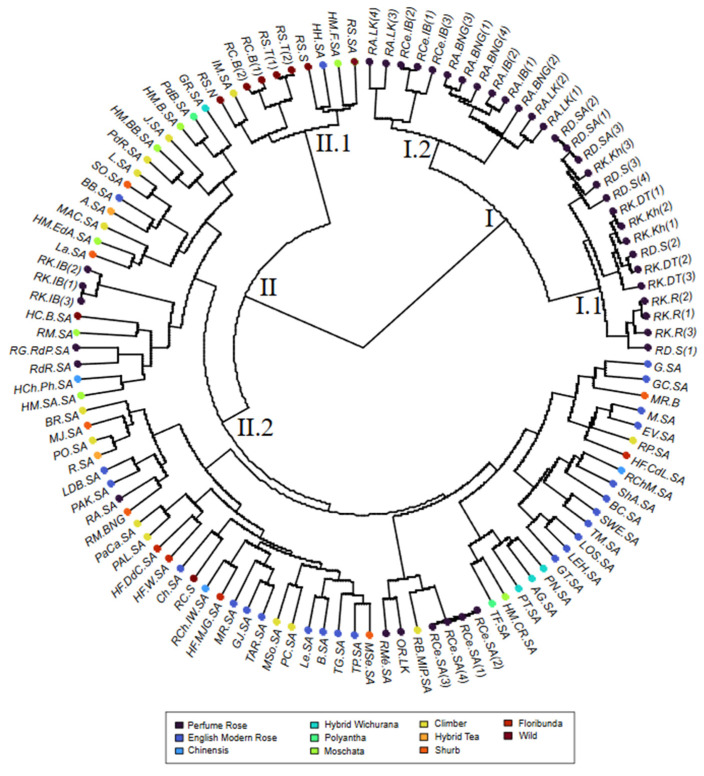
Hierarchical ascending classification of the 114 rose accessions based on Bruvo distance and Ward’s method (accession abbreviations are listed in Appendix A).

**Figure 2 plants-13-03563-f002:**
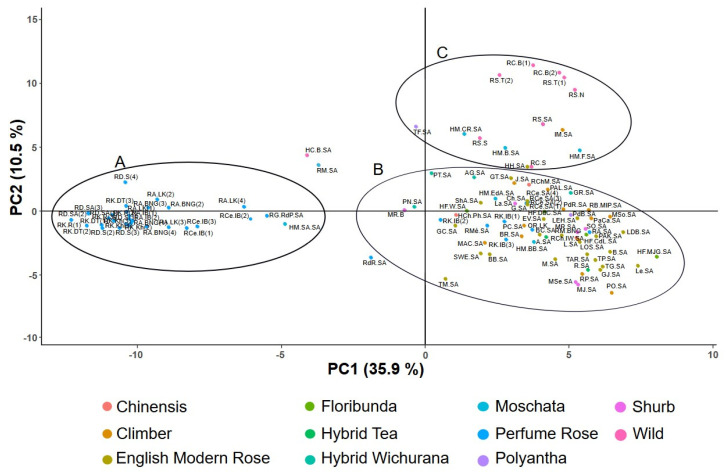
Principal component analysis of the 114 *Rosa* L. accessions based on the 7 SSR primers where (**A**–**C**) represent the three rose groups (accession abbreviations are listed in Appendix A).

**Figure 3 plants-13-03563-f003:**
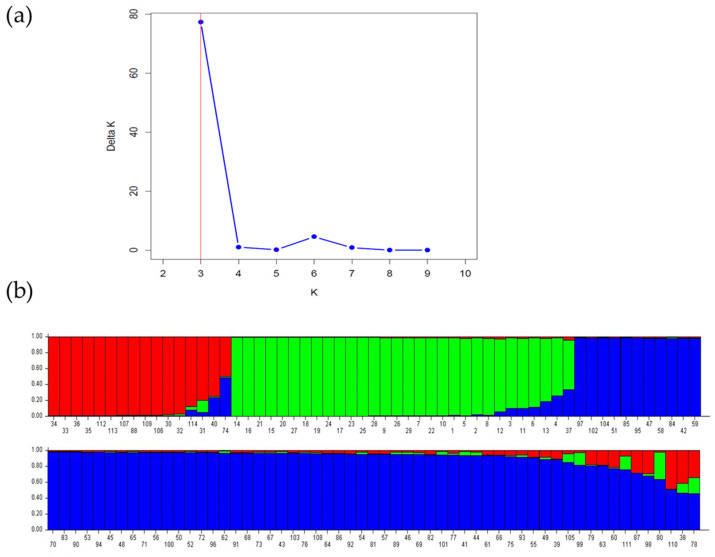
Genetic structure analysis of 114 rose accessions: (**a**) Evanno coefficient biplot displaying delta K and K values, the ad hoc measure of Evanno; (**b**) bar chart representing the 114 accessions organized according to their inferred ancestry at K = 3. Each individual is represented by a single vertical line, which is partitioned into colored segments in proportion with the estimated membership in the K clusters (code numbers are listed in Appendix A).

**Figure 4 plants-13-03563-f004:**
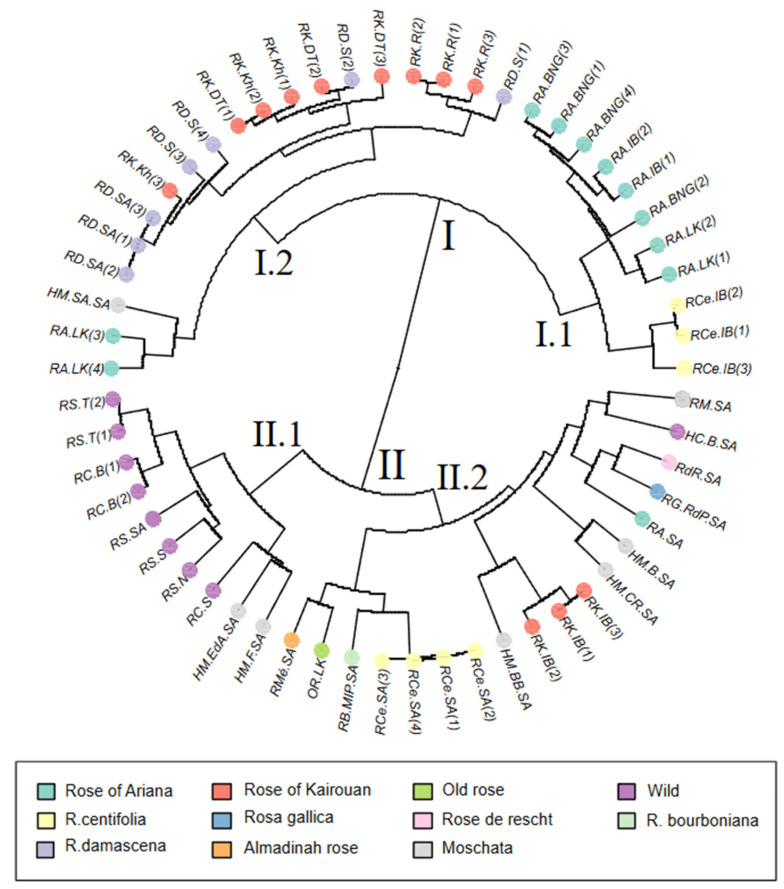
Hierarchical ascending classification including the perfumed roses accessions, structured with the Bruvo distance according to Ward’s method of classification (accession abbreviations are listed in Appendix A).

**Figure 5 plants-13-03563-f005:**
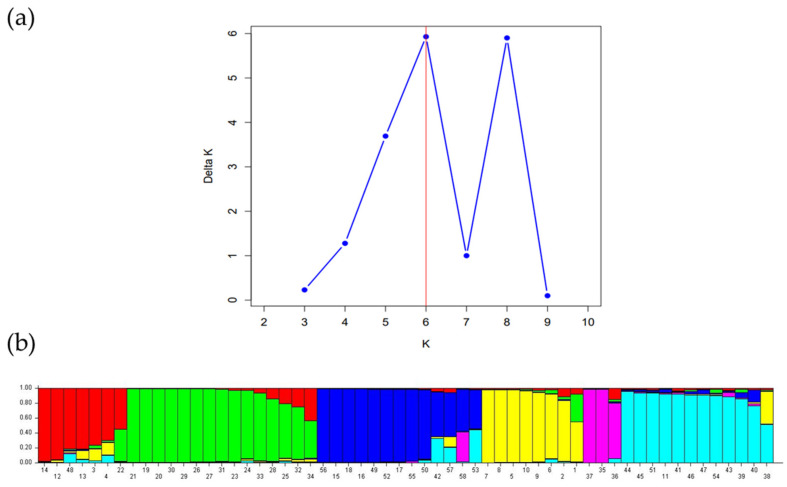
Genetic structure analysis of 58 perfumed rose accessions: (**a**) Evanno coefficient biplot displaying delta K and K values, the ad hoc measure of Evanno; (**b**) bar chart representing the 58 accessions organized according to their inferred ancestry at K = 6. Each individual is represented by a single vertical line, which is partitioned into colored segments in proportion to the estimated membership in the K clusters (code number are listed in Appendix A).

**Figure 6 plants-13-03563-f006:**
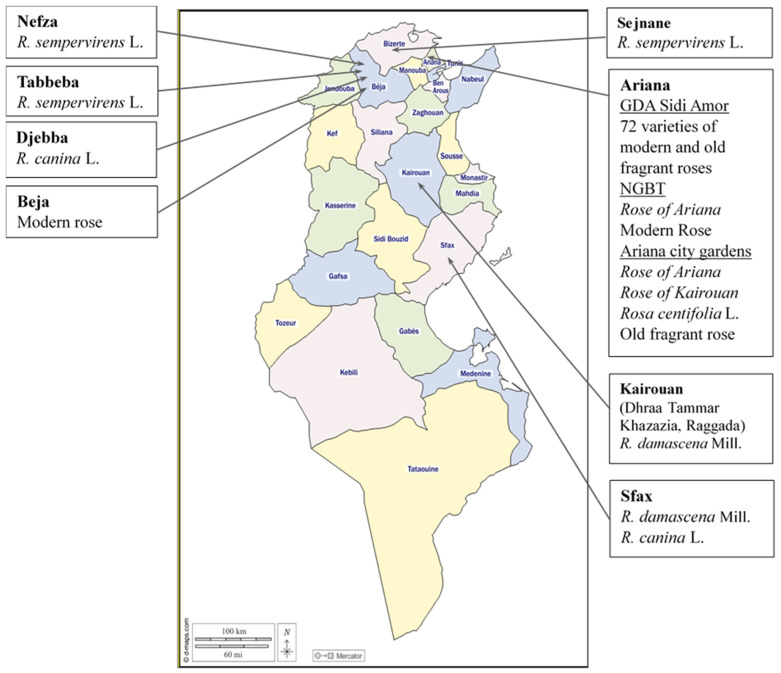
Surveyed areas and collected species of the *Rosa* L. genus in Tunisia.

**Table 1 plants-13-03563-t001:** Genetic parameters: number of alleles, size of bands (bp), expected heterozygosity (He), observed heterozygosity (Ho), polymorphism information content (PIC), and allelic phenotype number (APN) identified among the 114 *Rosa* accessions for the 7 SSR primers.

SSR	Number of Alleles	Size (bp)	He	Ho	PIC	APN
RhE2b	23	162–198	0.878	0.679	0.941	72
RW52D4	23	203–273	0.870	0.706	0.912	47
H10D03	22	205–246	0.924	0.583	0.927	55
RW10M24	22	253–292	0.894	0.220	0.911	40
RhD201	21	188–248	0.904	0.541	0.895	47
RhB303	19	119–152	0.803	0.527	0.881	54
H20D08	17	237–290	0.903	0.128	0.891	28
Average	21	195–243	0.882	0.483	0.908	
Total	147					343

**Table 2 plants-13-03563-t002:** Distribution of molecular variance analysis (AMOVA) among and within the 114 rose accession groups. Degrees of freedom (DF), sum of squares (SS), mean of squares (MS), estimated variance (EV), and percentage of variance (PV).

Source	DF	SS	MS	EV	PV	PhiPT ^1^	*p*-Value (Rand ≥ Data) ^2^
Among group	10	9.252	0.925	0.068	11%	0.180	0.001
Within group	103	31.717	0.308	0.308	89%		
Total	113	40.969		0.376	100%		

^1^ PhiPT: an analog of Wright’s FST for dominant binary data. ^2^ *p*-value: statistical significance.

**Table 3 plants-13-03563-t003:** Allelic phenotypes of SSR markers for *R. centifolia* L. (RCe.IB), *R. gallica*—‘Rose de Provins’ (RG.RdP.SA), and ‘Rose of Ariana’ accessions.

	**SSR Primers ^1^**						
**Rose Samples**	**H10D03**	**H20D08**	**RhB303**	**RhD201**	**RhE2b**	RW10M24	RW52D04
RA.LK(1)	(208, 223, 240)	258	129	203	(179, 184, 186)	284	(**204**, 218)
RA.LK(2)	(208, 223, 240)	260	127	203	(177, **180**, 186)	**285**	207
RA.LK(3)	(208, 223, 240)	258	128	**200**	(177, **182**, 186)	**285**	(**204**, **216**)
RA.LK(4)	(208, 225, 240)	265	129	**200**	(**175**, **182**, 186)	0	**204**
RA.BNG(1)	(**205**, 223, 240)	261	126	(**197**, 203)	(167, 176, **180**)	**285**	(**204**, 218)
RA.BNG(2)	(**205**, 220, 235)	**258**	126	203	(167, 176, **180**)	284	(207, 218)
RA.BNG(3)	(**205**, 223, **239**)	260	126	203	(165, 176, **180**)	**285**	(207, 218)
RA.BNG(4)	(208, 223, **239**)	254	130	(198, 203)	(167, 176, **180**)	**285**	(**204**, 218)
RA.IB(1)	(208, 225, 240)	260	126	(198, 203)	(165, 175)	**285**	(**204**, 218)
RA.IB(2)	0	260	126	(198, 203)	(175, **180**)	284	0
RCe.IB(1)	(**205**, 223, **239**)	260	(127, **135**)	(**197**, **200**)	(**175**, **180**, 187)	**285**	(**204**, **216**)
RCe.IB(2)	0	260	(127, **135**)	0	(176, **182**, 187)	0	0
RCe.IB(3)	(**205**, 223, **239**)	260	(127, **135**)	(**197**, **200**)	0	**285**	(**204**, **216**)
RG.RdP.SA	(223, 235)	(260, 290)	127	(198, 203, 215)	(167, 176, 187)	283	(207, 213)

^1^ Allele values in bold, derived from ‘*R. centifolia* L.’; the underlined values correspond to ‘*R. gallica*’—‘Rose de Provins’.

**Table 4 plants-13-03563-t004:** Allelic phenotypes of SSR markers for *R. centifolia* L. (RCe.SA), *R. bourboniana* Desp. (RB.MIP.SA), and ‘Rose of Kairouan’ (RK.IB).

	**SSR Primers ^1^**						
**Rose Samples**	**H10D03**	**H20D08**	**RhB303**	**RhD201**	**RhE2b**	RW10M24	RW52D04
RB.MIP.SA	**232**	**247**	(**131**, **147**)	**206**	(**167**, **188**)	**273**	(**216**, **231**)
RK.IB(1)	(223, **232**)	**247**	(127, **147**)	(197, **206**, 239)	(184, 193)	(253, 265)	(**216**, **231**)
RK.IB(2)	(223, **232**)	**247**	(127, **135**)	(199, **206**, 240)	(184, 193)	(253, 285)	(**216**, **231**)
RK.IB(3)	(223, **232**)	**247**	(127, **147**)	(197, **206**, 239)	(184, 193)	0	(**216**, **231**)
RCe.SA(1)	**232**	260	(119, 127, **147**)	(198, 203)	(**167**, **188**)	**273**	(212, **231**)
RCe.SA(2)	**232**	260	(119, 127, **147**)	(198, 203)	(**167**, **188**)	**273**	(212, **231**)
RCe.SA(3)	**232**	260	(119, 127, **147**)	(199, 203)	(**167**, **188**)	**273**	(212, **231**)
RCe.SA(4)	**232**	260	(119, 127, **147**)	(198, 203)	(**167**, **188**)	**273**	(212, **231**)

^1^ Allele values in bold derived from ‘*R. bourboniana* Desp.’.

**Table 5 plants-13-03563-t005:** List, forward and reverse sequences, linkage group (LG), and annealing temperature (Ta) of the 7 SSR primers.

SSRs	Forward (5′-3′)	Reverse (5′-3′)	LG	Ta	Ref.
RhD201	GGTATGCAAATAAGAGATACAGT	GTTTCTTCCTAACAAACCCATTTTGAAAGGG	1	53 °C	[47]
RhB303	CACTGCAACAACCCAATAGC	GTTTCTTGTCTTCAGCTTAGACTGTGCTG	2	50 °C	[47]
H20D08	TTCGGCTCTCTTCTCTGCTC	GACATTACAGCGACGAAGCA	4	53 °C	[19]
RW52D4	GGCAGTTGCTGTGCAGTG	TTGTGCCGACTCAAAATCAA	5	55 °C	[19]
RhE2b	CTTTGCATCAGAATCTGCTGCATT	GTTTCTTGCAGACACAGTTCATTAAAGCAG	6	53 °C	[47]
H10D03	CAATTCAAAACCACCGCTCT	CGCAGAGTCAACGAACCATA	7	55 °C	[19]
RW10M24	TTAATCCAAGGTCAAAGCTG	TCTCTTTCCCTCCTCACTCT	7	53 °C	[48]

## Data Availability

Data are contained within the article and Appendix A.

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
