# Peer review of "Genetic Diversity and Relationships Among Tunisian Wild and Cultivated Rosa L. Species"

_plants, 2024, doi:10.3390/plants13243563_

Round 1
Reviewer 1 Report
Comments and Suggestions for Authors
The presented manuscript contributes to the knowledge of the diversity and taxonomy of the genus Rosa.
There are comments that should be considered by the authors.
Lines 24-25: “most perfumed rose accessions (R. damascena and R. centifolia) formed a distinct group”.- No, they are not grouped together on trees.
Line 135 and following: What is meant by “genetic group” in this study?
Figures 1, 2, and 4 legends: add: “For accessions abbreviations refer to Table S1”.
In Figures 3 and 4, are the column numbers indicate the accession numbers? In the supplementary Table S1 the accessions should be numbered. Figure captions should include a reference to this table.
Line 334: the same cluster > the same cluster I (Figure 1)
Tables 3 and 4 Legends: add meaning of numbers in the tables.
Line 359: group > Cluster II
Line 365: R. damascena RD.R, RD.DT, and RD.KH are not studied in the current investigation.
Lines 369-371: Please check this sentence.
Lines 372-73: “consecutive groups” - what groups?
Lines 387-88: “identify the identity” - ?
Lines 394-407: It should be added that according to the HAC data in Figure 1, R. gallica RG.RdP.SA is far from ‘Rose of Ariana’ and is even located in a different cluster.
Line 406: “the proximity” – of what?
Line 512: pi2 > pi2
Author Response
December 14, 2024
Dear reviewer
Please find enclosed the revised manuscript entitled “Genetic Diversity and Relationships Among Tunisian Wild and Cultivated Rosa Species”, which would be accepted after minor revisions by the journal Plants.
We have carefully addressed the comments and suggestions provided. Below, we have outlined the revisions made in response to their feedback (all changes are highlighted in green in the manuscript):
Reviewer 1
1. Lines 24-25: “most perfumed rose accessions ( damascena and R. centifolia) formed a distinct group”. - No, they are not grouped together on trees.
Thank you for your valuable comment. We changed “One key finding was that most perfumed rose accessions (R. damascena and R. centifolia) formed a distinct group” for “One key finding was that most perfumed rose accessions (R. damascena and R. centifolia) are distinct from the other rose accessions, indicating a unique genetic pool for these roses.”
2. Line 135 and following: What is meant by “genetic group” in this study?
Thank you very much. The term "genetic group" was replaced by "group".
3. Figures 1, 2, and 4 legends: add: “For accessions abbreviations refer to Table S1”. Legends for Figures 1, 2, and 4.
Thank you very much for your suggestions. Changes have been made as requested by the reviewer.
4. In Figures 3 and 4, are the column numbers indicate the accession numbers? In the supplementary Table S1 the accessions should be numbered. Figure captions should include a reference to this table.
Thank you for your thorough review and valuable comments. The accession numbers of Figures 3 and 5 have been added in Supplementary Table S1.
5. Line 334: the same cluster > the same cluster I (Figure 1)
Changes have been made as requested by the reviewer.
6. Tables 3 and 4 Legends: add meaning of numbers in the tables.
Changes have been made as requested by the reviewer.
7. Line 359: group > Cluster II
Changes have been made as requested by the reviewer.
8. Line 365: R. damascena RD.R, RD.DT, and RD.KH are not studied in the current investigation.
My apologies, it was an error. RD.R, RD.DT, and RD.Kh was replaced by RK.R, RK.DT, and RK.Kh.
9. Lines 369-371: Please check this sentence.
Thank you very much, your suggestions have been taken into account. The sentence has been corrected as ‘Focusing on sub-cluster II.2 (Figure 1), which includes almost all hybrids of roses and the modern English roses of David Austin (except HM.F.SA, HH.SA, and IM.SA), was not grouped according to their species, or hybrid status, unlike other groups.’
10. Lines 372-73: “consecutive groups” - what groups?
Thank you very much for your appreciation. "Consecutive groups" has been changed to "rose groups" to refer to the groups listed in Supplementary Table S1 (e.g., English modern rose, hybrids, climbers, etc.).
11. Lines 387-88: “identify the identity” ?
I agree, it is confusing. "Identify the identity" has been replaced with "reveal the identity."
12. Lines 394-407: It should be added that according to the HAC data in Figure 1, R. gallica RG.RdP.SA is far from ‘Rose of Ariana’ and is even located in a different cluster.
Changes have been made as requested by the reviewer.
13. Line 406: “the proximity” – of what?
Thank you very much again for your suggestions to improve the clarity of the manuscript. The proximity of the 'Rose of Ariana' to R. centifolia has been clarified with a more precise formulation of the sentence.
14. Line 512: pi2 > pi2
Changes have been made as requested by the reviewer.
Thank you very much for your time in reviewing the manuscript and for your comments, which have undoubtedly helped to improve it. We believe these revisions have further strengthened the manuscript for publication in Plants journal.
Reviewer 2 Report
Comments and Suggestions for Authors
In the article entitled “Genetic diversity and relationships among Tunisian wild and cultivated Rosa species”, the authors conducted an investigation of genetic diversity among various rose species at different ploidy levels in Tunisia. The study was well-designed and conclusion was supported by the experimental data. However, the following concerns should be well-addressed before considering acceptance.
1. There are too many paragraphs in the introduction section and discussion section, and some small paragraphs should be merged.
2. The text in Figure 3 is not clear. Please replace Figure 3 with a higher resolution image.
3. All μl should be μL, and MgCl2 should be MgCl2 (subscript).
4. At least one representative agarose electrophoresis image of PCR products from SSR analysis should be provided.
5. The specific location (latitude and longitude) of the collection site for plant samples used for SSR analysis should be provided in the Materials and Methods section.
Author Response
December 14, 2024
Dear reviewer
Please find enclosed the revised manuscript entitled “Genetic Diversity and Relationships Among Tunisian Wild and Cultivated Rosa Species”, which would be accepted after minor revisions by the journal Plants.
We have carefully addressed the comments and suggestions provided. Below, we have outlined the revisions made in response to their feedback (all changes are highlighted in green in the manuscript):
Reviewer 2
1. There are too many paragraphs in the introduction section and discussion section, and some small paragraphs should be merged:
Thank you very much for your valuable suggestion. Changes have been made as requested by the reviewer.
2. The text in Figure 3 is not clear. Please replace Figure 3 with a higher-resolution image:
Thank you for providing your valuable feedback. Changes have been made as requested by the reviewer and we replaced Figure 5 too.
3. All μl should be μL, and MgCl2 should be MgCl2 (subscript):
Changes have been made as requested by the reviewer.
4. At least one representative agarose electrophoresis image of PCR products from SSR analysis should be provided.
You are right, it is important to provide evidence of the markers used. We added it in supplementary materials as Figure S1.
5. The specific location (latitude and longitude) of the collection site for plant samples used for SSR analysis should be provided in the Materials and Methods section.
Thank you very much for your valuable feedback. Changes have been made as requested by the reviewer.
Thank you very much for your time in reviewing the manuscript and for your comments, which have undoubtedly helped to improve it. We believe these revisions have further strengthened the manuscript for publication in Plants journal.